# Sample-Based Constrained Inference for Matrix-Free Quantum Process Tomography

## Abstract

Quantum process tomography reconstructs an unknown quantum channel from finite measurement counts. For sample-based uncertainty reporting, the sampled candidate channels should also remain physically valid, meaning completely positive and trace preserving (CPTP). Motivated by a flow-matching inverse-problem view, we introduce a matrix-free CPTP channel-sampling protocol for simulated full-basis process tomography. The protocol stores repeated tomography basis objects by index, constructs candidate channels with normalized Kraus factors, and calibrates scalar intervals using records from systems with one or two qubits before applying the fixed rule to four-qubit records. On a four-qubit controlled-NOT-family process, the center reconstruction reaches process infidelity 0.0143, average gate fidelity 0.9865, and passes projected CPTP diagnostics. Raw four-qubit intervals are under-dispersed, covering 0 out of 12 held-out scalar records, while the lower-qubit-selected calibrated intervals cover 12 out of 12 records at nominal 90% level with mean width 0.0338. The calibrated four-qubit run stays within a declared 1800 second and 8192 MB resource envelope, taking 1287.49 seconds and 4000.04 MB. These results show physically valid four-qubit reconstruction with lower-qubit-calibrated scalar intervals under fixed acquisition in a controlled simulator setting.

## 1 Introduction

Quantum process tomography estimates the input-output action of a quantum device from preparations, measurements, and finite counts. The object of interest is a quantum channel, and a valid channel must be completely positive and trace preserving (CPTP) (Choi, 1975; Chuang & Nielsen, 1997; Poyatos et al., 1997). A reconstruction method for this setting must fit the observed counts while keeping the reconstructed channel inside the physically meaningful set of quantum dynamics.

Sample-based uncertainty reporting makes the physicality requirement stricter than it is for a single point estimate. A nonphysical point estimate can sometimes be diagnosed or projected after fitting, but uncertainty over realizable dynamics is interpretable only when the sampled channels are themselves physical. We report a center channel together with conditional channel samples and scalar intervals, so the sample construction, diagnostics, and interval reporting must all respect the CPTP constraint.

The second difficulty is the scaling of full process tomography. We use the Choi matrix as the matrix representation of a quantum channel. For $N$ qubits, the input dimension is $d = 2^N$, the Choi matrix has size $d^2 \times d^2$, and a dense Hermitian parameterization has $d^4$ real degrees of freedom. The four-qubit setting already has a $256 \times 256$ Choi matrix and 65,536 Hermitian parameters. In the recorded four-qubit full-basis experiment used in this paper, a dense design matrix would contain 331,776 observation rows and more than $2.17 \times 10^{10}$ entries. This size is small compared with the Hilbert spaces that appear in larger quantum systems, but it is already large enough that materializing the dense design is the wrong abstraction for uncertainty-aware channel sampling.

Measurement-efficient approaches such as classical-shadow process tomography attack a different part of the problem by changing the measurement design (Levy et al., 2024). Our setting is complementary: finite-count tomography records are already given, and the reconstruction method must turn those records into physically

valid channel samples and calibrated scalar summaries. We study this reconstruction side under simulated full-basis process tomography, where simulator truth is available for fidelity and interval evaluation.

Optimization-based QPT has also shown that operator-constrained parameterizations can keep channel reconstruction inside the physical constraint set: Kraus parameterizations represent the channel action through matrix operators, and Stiefel parameterizations place those operators on an orthogonality-constrained matrix set used to enforce trace preservation (Ahmed et al., 2023; Quiroga & Kyrillidis, 2023; Volya et al., 2024). We build on this constrained-representation viewpoint, but target a different output: a fixed finite-count record is converted into matrix-free channel samples and scalar intervals, with calibration selected on lower-qubit records before four-qubit evaluation.

Generative inverse-problem methods, including diffusion and flow-matching solvers, motivate the way we pose this reconstruction problem (Chung et al., 2023; Lipman et al., 2023; Tong et al., 2024; Pourya et al., 2026). An observation map sends a latent object to predicted measurements; inference returns samples that explain the observations while remaining in the valid target set. Flower is the closest motivation for our framing: it combines a flow-based generative model with measurement-consistent refinement for linear inverse problems (Pourya et al., 2026). We do not train or use a neural flow-matching solver. Instead, we use the inverse-problem decomposition into an observation map, feasible target set, conditional sample construction, and calibration audit for QPT under a CPTP constraint. The technical problem is to combine these pieces while keeping the four-qubit design matrix implicit and selecting uncertainty calibration rules before four-qubit truth is opened for evaluation.

This paper presents a matrix-free CPTP channel-sampling protocol for finite-count quantum process tomography. The first component is an indexed observation adapter that converts Qiskit process-tomography records into prediction, residual, and adjoint interfaces while storing repeated basis objects by index (Javadi-Abhari et al., 2024; Qiskit Experiments Developers, 2026). The second component uses low-rank Kraus factors, matrices whose normalized products define a physical channel, to construct CPTP Choi matrices by design. The third component is a scalar interval calibration rule selected only on lower-qubit records and then applied without adaptation to four-qubit records. The same channel representation carries reconstruction, sample physicality, and interval reporting.

We evaluate the protocol on a simulated four-qubit controlled-NOT process. The center reconstruction reaches process infidelity 0.0143 and average gate fidelity 0.9865 while passing projected CPTP diagnostics. For scalar interval uncertainty, raw four-qubit quantile intervals cover 0 / 12 scalar records, while the lower-qubit-selected interval expansion covers all 12 at nominal 90% level. The calibrated interval run uses a declared 1800 second and 8192 MB resource envelope and records 1287.49 seconds and 4000.04 MB. The reported result is four-qubit simulator reconstruction with fixed-rule scalar interval coverage under a stated resource envelope.

Our contributions are as follows.

- We introduce a matrix-free CPTP channel-sampling protocol that combines an indexed observation cache with Kraus-factor channel samples, avoiding dense four-qubit design materialization.

- We separate center-sample reconstruction from scalar interval uncertainty, so the four-qubit point reconstruction claim is evaluated independently of calibration evidence.

- We report a non-leaking scalar interval protocol: the calibration rule is selected on records from systems with one or two qubits, then evaluated on four-qubit scalar records using simulator truth only after the rule is fixed.

- We provide an evidence-bounded four-qubit simulator study with physicality, fidelity, coverage, runtime, and memory measurements.

## 2 Background and Problem Setup

Before introducing notation, the data can be read as a table of repeated trials. Each row prepares a known input state, applies the unknown process, measures one output event, and records how often that event

occurred. The reconstruction problem is to turn this finite-count table into candidate quantum channels, where a channel is the input-output map being learned. A channel sample is one candidate map conditioned on the same table. The interval results in this paper are not intervals over the entire channel; they are intervals over selected scalar summaries of the sampled channels.

For readers who prefer an inverse-problem view, Appendix A translates the quantum notation below into the roles of known inputs, measurement queries, candidate maps, feasibility constraints, and scalar uncertainty summaries.

Let $N$ denote the number of qubits and let $d = 2^N$ denote the Hilbert-space dimension of the input system. A quantum channel $\Phi$ is represented by its Choi matrix $\mathrm{J}(\Phi) \in \mathbb{C}^{d^2 \times d^2}$. We write the physically valid set as

$$\mathcal{C}_{\mathrm{CPTP}} = \left\{ \mathrm{J} \in \mathbb{C}^{d^2 \times d^2} : \mathrm{J} = \mathrm{J}^\dagger, \ \mathrm{J} \succeq 0, \ \mathrm{Tr}_{\mathrm{out}}(\mathrm{J}) = I_d \right\}, \tag{1}$$

where $\mathrm{J}^\dagger$ is the conjugate transpose of $\mathrm{J}$, $\mathrm{J} \succeq 0$ means that $\mathrm{J}$ is positive semidefinite, $\mathrm{Tr}_{\mathrm{out}}$ is the partial trace over the output subsystem, and $I_d$ is the $d \times d$ identity matrix. The first two constraints encode complete positivity, and the partial-trace constraint encodes trace preservation (Choi, 1975).

Each tomography observation row is defined by an input density matrix $\rho_r \in \mathbb{C}^{d \times d}$, a measurement effect $M_r \in \mathbb{C}^{d \times d}$, a shot count $m_r$, and an observed count $y_r$, where $r$ indexes the row. The empirical probability is $\hat{p}_r = y_r/m_r$. For a candidate channel $\Phi$, the predicted probability is

$$p_r(\Phi) = \mathrm{Tr}[M_r \Phi(\rho_r)]. \tag{2}$$

Equivalently, after converting the row into a Choi-space operator $A_r$, the same prediction can be written as $p_r(\mathrm{J}) = \mathrm{Tr}(A_r \mathrm{J})$. The dense diagnostic path uses $A_r = \rho_r^T \otimes M_r$, matching the column-vectorized Choi convention used by the implementation. The observation adapter provides the vector $H(\mathrm{J}) = (p_r(\mathrm{J}))_r$, the residual $H(\mathrm{J}) - \hat{p}$, and the adjoint $H^*$, where $\hat{p}$ is the vector of empirical probabilities.

The reconstruction objective used by the protocol is the squared probability residual

$$L(\mathrm{J}) = \sum_r \left( p_r(\mathrm{J}) - \hat{p}_r \right)^2. \tag{3}$$

At four qubits, the method evaluates this objective through a cache of unique input density matrices and measurement effects, which keeps repeated basis rows indexed rather than materializing the full dense design. The paper-facing simulator records use a fixed shot count per circuit, so this unweighted probability residual is the fixed-shot fitting convention used by the reported implementation rather than a claim of optimal binomial weighting.

The four-qubit finite-count record has 20,736 circuits, 256 shots per circuit, and 331,776 observation rows. Its structured cache contains 256 unique input states and 1296 unique measurement effects. This controlled simulator record provides target-channel truth for evaluating reconstruction fidelity and scalar interval coverage.

Table 1: Scaling quantities used to motivate the structured observation interface. The four-qubit row is the paper-facing simulator setting; dense estimates provide resource context for the representation choice.

| Setting | Circuits | Observation rows | Choi side | Hermitian parameters |
|---|---|---|---|---|
| 1 qubit full basis | 12 | – | 4 | 16 |
| 2 qubit full basis | 144 | 576 | 16 | 256 |
| 4 qubit full basis | 20,736 | 331,776 | 256 | 65,536 |

The main scalar metrics are fidelity, physicality, and interval coverage. Process fidelity $F_{\mathrm{proc}}$ is computed against simulator truth using the Qiskit channel-fidelity convention with CPTP checks disabled during metric evaluation; fidelity is therefore an accuracy number, not a proof that the candidate channel is physical.

Physicality is reported separately through projected CPTP diagnostics: the reported Choi matrix is checked for Hermitian symmetry, positive semidefiniteness, and trace preservation after any stated channel projection or Kraus normalization step. Average gate fidelity is derived from $F_{\text{proc}}$ as

$$F_{\text{avg}} = \frac{dF_{\text{proc}} + 1}{d + 1}. \tag{4}$$

Process infidelity is $1 - F_{\text{proc}}$. For an interval $[\ell_s, u_s]$ associated with a scalar $s$, coverage means that the simulator-truth scalar value lies inside the interval. The nominal level used in the interval records is 90%.

The word posterior is used in a computational sense in this paper. A posterior sample means a CPTP channel sample generated conditionally on the fixed finite-count tomography record by perturbing a reconstructed Kraus center and re-normalizing the Kraus factors. This is not a claim that the samples are draws from a fully specified Bayesian posterior distribution. The reported uncertainty object is a scalar interval derived from these conditional CPTP samples and, when stated, expanded by a lower-qubit-selected calibration rule.

## 3 Method

### 3.1 Overview

Our protocol couples three components that are usually audited separately: a tomography observation adapter, a low-rank CPTP channel parameterization, and a scalar interval calibration layer. Figure 1 summarizes the execution order. Given finite-count process-tomography data, the observation adapter builds an indexed residual interface. The channel module initializes or perturbs Kraus factors and converts them into Choi matrices that are CPTP by construction. The uncertainty module forms scalar intervals from perturbed Kraus samples and applies a lower-qubit-selected calibration rule before evaluating four-qubit coverage.

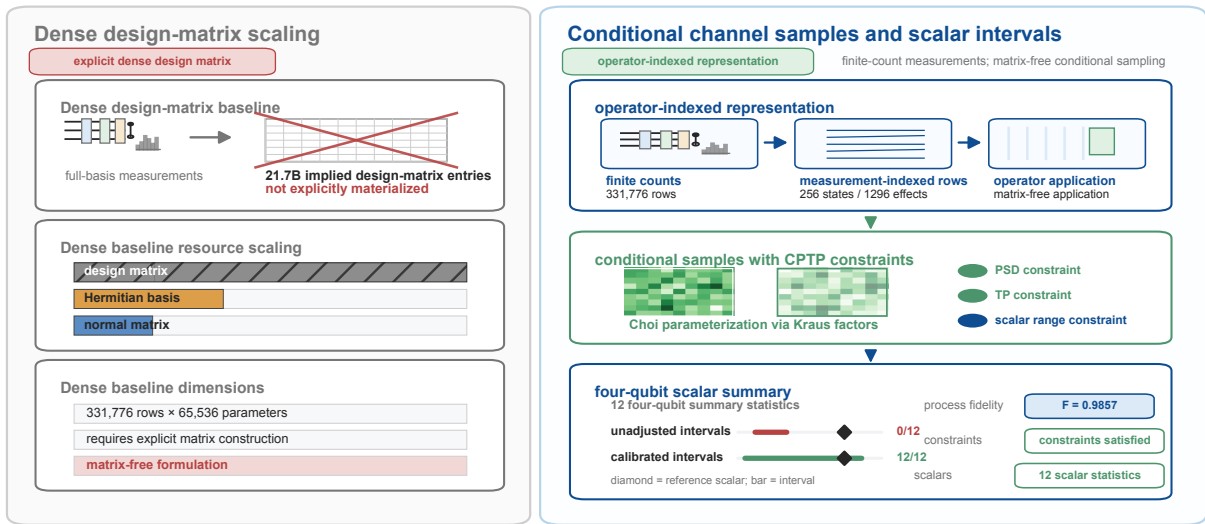

Figure 1: Overview of the reported simulator study. Left: direct dense materialization of the full four-qubit Choi design gives the dense baseline. Right: the structured path keeps finite-count tomography records indexed, constructs CPTP channels through Kraus factors, and separates CPTP samples from scalar interval reporting. The calibration rule is selected on lower-qubit records and then applied to four-qubit records without using four-qubit truth for rule selection.

**Execution contract.** The reported run uses a fixed execution order. First, the finite-count tomography record is converted into an indexed cache of preparations, effects, outcomes, counts, and shots. Second, the

center path constructs a rank-1 Kraus channel from the adjoint-spectral initializer and evaluates fidelity and CPTP diagnostics against simulator truth. Third, the uncertainty path perturbs the same Kraus center, renormalizes each sample to satisfy trace preservation, converts each sample to a Choi matrix, and forms raw scalar intervals. Fourth, the lower-qubit-selected calibration margins are applied to those scalar intervals. Four-qubit truth is used only in the final coverage evaluation, after the calibration rule is fixed. The four-qubit computation uses implicit design-matrix evaluation and deterministic Kraus perturbations throughout the reported protocol.

## 3.2 Indexed Observation Adapter

The observation adapter is needed because dense design materialization is the dominant avoidable representation cost. For each circuit row, the adapter stores the preparation index, measurement index, outcome, count, and shot count. For four-qubit data, basis payloads are stored as indices rather than dense matrices during serialization; dense input states and measurement effects are reconstructed only as unique cached arrays. The cache records row count, unique input count, unique effect count, empirical probabilities, and observed counts. In the four-qubit record, this turns 331,776 rows into indexed references to 256 input states and 1296 measurement effects.

The cached prediction computes row probabilities without looping over every dense Choi design coefficient. Let $K_1, \ldots, K_R$ be Kraus matrices for a candidate channel, where $R$ is the chosen Kraus rank. For each unique input state $\rho$, the adapter evaluates

$$\Phi_K(\rho) = \sum_{a=1}^{R} K_a \rho K_a^\dagger. \tag{5}$$

It then contracts the evolved states with unique measurement effects and indexes the resulting probability table back to observation rows. This produces the residuals in equation 3 while avoiding the dense Hermitian design matrix that would dominate the four-qubit memory footprint.

In the recorded four-qubit center run, the cache is the representation mechanism that turns 331,776 observation rows into lookups over 256 unique input states and 1296 unique measurement effects. The same record estimates the dense alternative at 21,743,271,936 design entries, with 165,888 MB for the dense design matrix, 65,536 MB for the Hermitian basis, and 32,768 MB for the normal matrix. These quantities motivate the indexed interface and matrix-free execution path.

The adapter also fixes the convention boundary between quantum data generation and numerical reconstruction. Qiskit supplies the process-tomography circuits and channel objects, while the reconstruction protocol receives only the row-level objects needed for prediction, residuals, and adjoints (Javadi-Abhari et al., 2024; Qiskit Experiments Developers, 2026). This boundary localizes failures: if a metric or physicality diagnostic fails, the source can be traced to the channel representation, the observation convention, or the metric convention instead of a monolithic fitting routine.

## 3.3 CPTP-by-Construction Channel Samples

Conditional channel samples require physicality inside the sampling construction. We represent a candidate channel with Kraus matrices $K_1, \ldots, K_R \in \mathbb{C}^{d \times d}$. The trace-preserving condition is

$$\sum_{a=1}^{R} K_a^\dagger K_a = I_d. \tag{6}$$

Given arbitrary complex blocks, the implementation enforces this condition by computing the Gram matrix $G = \sum_a K_a^\dagger K_a$, taking a Hermitian inverse square root $G^{-1/2}$, and replacing each block by $K_a G^{-1/2}$. The corresponding Choi matrix is

$$\mathrm{J}(K_1, \ldots, K_R) = \sum_{a=1}^{R} \mathrm{vec}(K_a) \, \mathrm{vec}(K_a)^\dagger, \tag{7}$$

where $\mathrm{vec}(K_a)$ stacks the columns of $K_a$. This construction produces a positive-semidefinite matrix, and the normalization enforces trace preservation.

The reported center sample uses rank $R = 1$, adjoint-spectral initialization, one restart, and the probability-residual objective. Rank $R = 1$ is the center representation used for the noiseless controlled-NOT simulator row: a unitary channel admits a single Kraus operator, so the rank-1 center tests whether the indexed observation interface can recover a compact physical channel in this controlled setting. The adjoint-spectral initializer computes $H^*(\hat{p})$, symmetrizes it, extracts positive eigendirections, reshapes them into Kraus blocks, and applies the trace-preserving normalization. Intuitively, this back-projects the empirical probabilities through the observation adjoint and keeps the leading positive channel-shaped direction. In the reported center path, optimizer steps are set to zero after initialization, so the four-qubit center record is a deterministic structured reconstruction rather than an optimized dense least-squares solution.

In this representation, physicality is built into the reported samples. The Choi matrix in equation 7 is a sum of outer products and is positive semidefinite up to numerical precision. The normalization in equation 6 enforces trace preservation before metrics are computed. The experiments still report Hermitian, positive-semidefinite, trace-preserving, and CPTP diagnostics, but those diagnostics audit a by-construction sample rather than rescue an unconstrained one.

### 3.4 Center Reconstruction Path

The center reconstruction path is separated from scalar interval uncertainty because the two claims require different evidence. The center path asks whether the structured CPTP representation and indexed residual interface can reconstruct the target channel accurately enough at four qubits. It uses the deterministic adjoint-spectral initialization, rank $R = 1$, noise scale 0.0, one restart, and the probability-residual objective in equation 3. The output is a single CPTP Choi matrix whose process infidelity, average gate fidelity, and physicality diagnostics are measured against simulator truth.

Separating the center path prevents an uncertainty result from masking a reconstruction result. A calibrated interval can cover truth even when the center is poor if the interval is wide enough, and a strong center estimate can coexist with under-dispersed raw sample intervals. We therefore report the four-qubit center reconstruction and the calibrated scalar interval evidence as distinct results.

### 3.5 Conditional Samples and Scalar Intervals

Under the posterior definition above, conditional samples are generated by perturbing the fitted Kraus center and re-normalizing each perturbed block set back to the trace-preserving manifold. The reported four-qubit interval-sample configuration uses rank $R = 1$, noise scale 0.01, four samples per repetition, and the same probability-residual convention as the center path. For each sample, the implementation draws independent complex Gaussian noise with real and imaginary parts distributed as $\mathcal{N}(0, 0.01^2)$, adds it to the fitted Kraus blocks, and then applies the trace-preserving normalization in equation 6. After normalization, each sampled channel is converted to a Choi matrix and checked for CPTP diagnostics before scalar summaries are formed.

Let $s(\mathrm{J})$ be a scalar statistic of a sampled Choi matrix, such as process fidelity or a real Choi entry. For each scalar $s$, the raw interval is the empirical central 90% interval $[\ell_s, u_s]$ from sampled values. The recorded scalar set contains process fidelity, $\mathrm{Re}(\mathrm{J}_{0,0})$, and $\mathrm{Re}(\mathrm{J}_{1,1})$. Rule selection uses eight lower-qubit records for each scalar. Evaluation then uses four held-out four-qubit records for each of the same three scalars, giving 12 four-qubit scalar records in total. These three scalar summaries are the reported uncertainty quantities used for the fixed-record coverage audit.

The raw intervals are calibrated with scalar-specific margins selected on lower-qubit records. For each scalar $s$, the selected rule has a lower expansion $b_s \geq 0$, an upper expansion $a_s \geq 0$, and a small numerical tolerance $\epsilon = 10^{-9}$. The calibrated interval is

$$[\ell_s - b_s - \epsilon, \ u_s + a_s + \epsilon]. \tag{8}$$

The selected rule uses safety multiplier 2.0 and is fit on records from systems with one or two qubits. For each scalar, the lower expansion is twice the largest lower-side miss on the lower-qubit records plus $\epsilon$, and

the upper expansion is twice the largest upper-side miss plus $\epsilon$. For process fidelity, the upper expansion is 0.054017 and the lower expansion is $10^{-9}$. For $\text{Re}(J_{0,0})$, the lower and upper expansions are 0.027995 and 0.003637. For $\text{Re}(J_{1,1})$, the lower and upper expansions are 0.003637 and 0.009639. Four-qubit truth is used only after this rule is fixed to evaluate whether the calibrated intervals cover the simulator-truth scalar values.

### 3.6 Resource-Aware Execution

The resource-aware execution path records what is materialized and what is streamed or cached. The four-qubit reported run uses the structured path: repeated inputs and effects are kept in indexed caches, Kraus actions are applied to unique inputs, and row probabilities are reconstructed by indexing contractions back to the row list.

Runtime and memory are treated as measured protocol quantities rather than informal implementation notes. The center reconstruction records stage timing, including data generation, observation construction, cache construction, objective evaluation, diagnostics, metrics, and artifact writing. The calibrated interval run has a declared resource envelope of 1800 seconds and 8192 MB, and the observed run reports 1287.49 seconds and 4000.04 MB.

## 4 Experiments

### 4.1 Protocol

All experiments use simulated Qiskit full-basis process tomography. The one-qubit records cover identity, depolarizing, and amplitude-damping channels. The two-qubit record uses a controlled-NOT process with depolarizing noise parameter $p = 0.02$. The four-qubit record uses the controlled-NOT family with $p = 0.0$, 20,736 circuits, 256 shots per circuit, and 331,776 observation rows. In this four-qubit simulator circuit, controlled-NOT gates act on qubits 0 to 1 and 2 to 3; $p = 0.0$ means that no depolarizing channel is applied after the ideal gates.

The experiments test two main claims. The center-sample claim asks whether the structured CPTP representation can reconstruct the four-qubit target with in-range fidelity metrics while passing projected CPTP diagnostics. The scalar-interval claim asks whether a calibration rule selected on lower-qubit records can cover four-qubit scalar truth after the rule is fixed. Diagnostic baselines and dense resource estimates clarify row roles and conventions.

Table 2: Paper-facing protocol. The lower-qubit records are used for rule selection and convention checks; the four-qubit record is the main simulator benchmark.

| Record | Channel family | Noise | Role | Paper-facing use |
|--------|----------------|-------|------|------------------|
| 1 qubit | identity; depolarizing; amplitude damping | mixed | calibration | rule selection |
| 2 qubits | controlled-NOT with depolarizing noise | $p = 0.02$ | calibration | rule selection |
| 4 qubits | controlled-NOT family | $p = 0.0$ | evaluation | center and interval evidence |

Metrics are computed against simulator truth. Process fidelity is computed through Qiskit's channel fidelity convention with complete-positivity and trace-preservation requirements disabled for metric evaluation; the physicality claim is enforced separately through the CPTP diagnostics. We report process infidelity $1 - F_{\text{proc}}$, average gate fidelity from equation 4, projected CPTP diagnostic status, scalar interval coverage, runtime, and peak resident memory. Diagnostic rows are reported separately from the main center and interval rows.

### 4.2 Four-Qubit Center Reconstruction

The center reconstruction satisfies the reported physicality diagnostics and fidelity-range checks on the four-qubit benchmark. Figure 2 summarizes the center reconstruction, interval coverage, and declared resource

use. The structured sampler constructs a CPTP Choi matrix by design, passes the projected CPTP diagnostics, and reaches process infidelity 0.0143 on the recorded four-qubit controlled-NOT process.

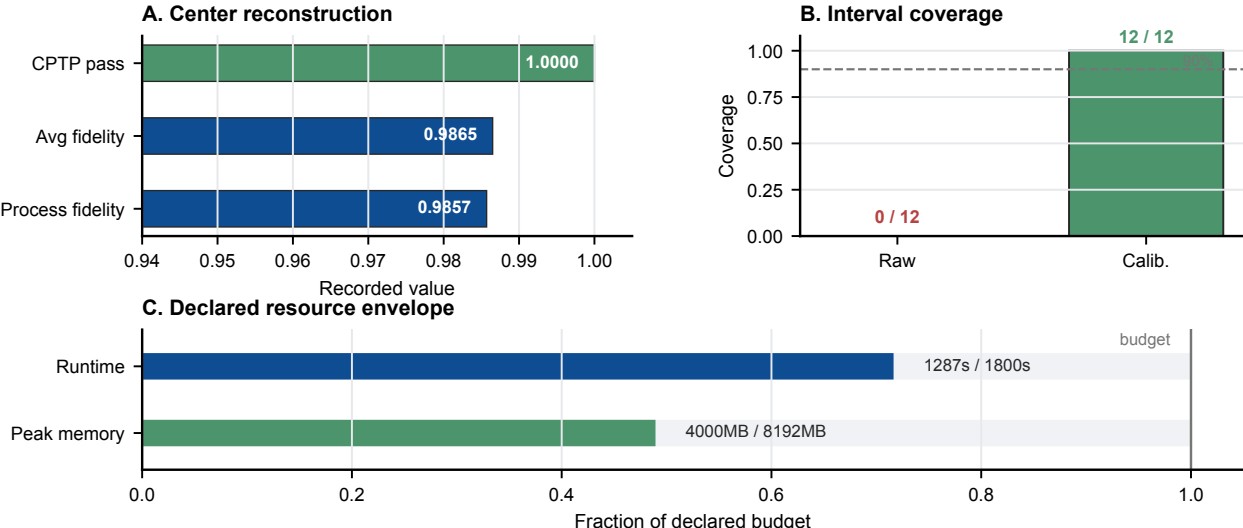

Figure 2: Main simulator evidence. Panel A reports the CPTP center reconstruction for the four-qubit controlled-NOT process using full-basis simulated process tomography with 20,736 circuits, 256 shots per circuit, and 331,776 observation rows; the CPTP label is a physicality diagnostic reported separately from the fidelity value. Panel B shows that raw four-qubit sample quantiles miss all scalar records, while the lower-qubit-selected interval rule covers all 12 held-out four-qubit scalar records. Panel C reports the declared resource envelope used by the calibrated interval run.

The center result is the reported reconstruction row. The output channel is CPTP by construction, passes projected CPTP diagnostics, and has no fidelity-range warning. The process fidelity is 0.9857, so the process infidelity is 0.0143. The average gate fidelity is 0.9865. These values are computed against the simulator target for the controlled four-qubit process in Table 2.

The dense resource estimate explains why the four-qubit experiment uses the indexed structured path. A dense Hermitian design would require more than $2.17 \times 10^{10}$ design entries for this setting, making dense materialization the representation cost that indexed evaluation avoids. Appendix D details the corresponding memory estimate.

### 4.3 Calibrated Scalar Intervals

The scalar-interval evidence separates raw sample variability from calibrated coverage. Raw four-qubit intervals from four conditional samples per record cover 0 / 12 scalar records. After applying the lower-qubit-selected scalar margins in equation 8, the calibrated intervals cover all 12 records at nominal 90% level. This is a fixed-record coverage audit, not a population-level coverage guarantee. Table 3 reports the per-scalar coverage and mean calibrated width.

Table 3: Four-qubit scalar interval coverage after applying the lower-qubit-selected calibration rule. Each scalar has four four-qubit records; the rule is selected without four-qubit truth.

| Scalar | Covered / total | Coverage | Mean width | Status |
|---|---|---|---|---|
| Process fidelity | 4 / 4 | 1.00 | 0.0546 | covered |
| $Re(J_{0,0})$ | 4 / 4 | 1.00 | 0.0334 | covered |
| $Re(J_{1,1})$ | 4 / 4 | 1.00 | 0.0134 | covered |
| All scalars | 12 / 12 | 1.00 | 0.0338 | covered |

The coverage table answers the empirical question used by the protocol: whether the fixed lower-qubit rule covers the selected four-qubit scalar records after it is applied without four-qubit tuning. The answer is yes for the recorded scalars.

The calibrated interval run also satisfies the declared resource envelope for the four-qubit uncertainty experiment. Table 4 shows the declared budget and observed usage. The run uses 16 distinct conditional channel samples across the recorded four-qubit scalar records; all pass projected CPTP diagnostics, and the sample-diversity check records a non-collapsed sample set. Its associated interval-run center has process infidelity 0.0169; this is distinct from the center-reconstruction row in Figure 2, which reports process infidelity 0.0143. These values support the fixed-rule calibrated-interval claim.

Table 4: Resource record for the calibrated four-qubit interval run. The declared budget is part of the experimental protocol; observed runtime and memory are measured in the recorded run.

| Experiment | Time budget | Time used | Memory budget | Memory used |
|---|---|---|---|---|
| Calibrated four-qubit interval | 1800 s | 1287.49 s | 8192 MB | 4000.04 MB |

### 4.4 Diagnostic Comparisons

Two diagnostic comparisons clarify the roles of the reported rows. First, the raw sample intervals have coverage 0 / 12, with mean raw width 0.000817. The calibrated intervals measure the effect of the lower-qubit-selected expansion rule on those same scalar records. Second, non-CPTP linear-inversion outputs can produce fidelity values outside the physical range; such rows audit metric conventions and are reported separately from the CPTP channel-sampling results.

Table 5: Roles of the main experimental and diagnostic rows.

| Row | Role | Paper-facing? | Interpretation |
|---|---|---|---|
| 4q center sample | reconstruction | yes | supports 4q center reconstruction |
| Calibrated 4q intervals | uncertainty | yes | supports fixed-rule scalar coverage |
| Raw 4q quantile intervals | diagnostic | no | shows under-dispersion before calibration |
| Non-CPTP linear inversion | diagnostic | no | audits metric conventions |
| Dense design resource estimate | context | no | quantifies dense representation cost |

These rows establish the two comparisons that the current evidence directly supports. The first comparison is representational: dense materialization exposes a large design-matrix cost, while indexed evaluation keeps the four-qubit record usable for channel sampling. The second comparison is statistical: raw sample intervals are under-dispersed on the recorded scalars, while the lower-qubit-calibrated intervals cover the same held-out scalar records.

The scalar-interval result uses a non-leaking lower-qubit expansion rule that is fixed before four-qubit evaluation. Once selected, this rule covers the recorded four-qubit scalar records under the declared resource envelope.

## 5 Related Work

Quantum process tomography provides the standard setting for reconstructing unknown quantum dynamics from preparations and measurements (Chuang & Nielsen, 1997; Poyatos et al., 1997). Classical full-basis tomography is conceptually direct because it probes a complete set of inputs and measurement outcomes, but the number of circuits and the size of the channel representation grow exponentially with the number of qubits. The present work stays in this full-basis simulator setting so that target channels and scalar truth values are available for auditing reconstruction and interval coverage.

Optimization-based QPT has already developed several physically constrained low-rank reconstruction strategies. GD-QPT learns Kraus operators directly and enforces trace preservation through optimization on a Stiefel manifold, an orthogonality-constrained matrix set (Ahmed et al., 2023); factored non-convex QPT uses a Burer–Monteiro-style low-rank process factorization for near-unitary gates (Quiroga & Kyrillidis, 2023); and Riemannian-QPT performs stochastic gradient updates on a Kraus/Stiefel representation (Volya et al., 2024). These methods are the closest prior work to the channel representation used here. This work fixes the finite-count observation record and couples an indexed residual interface with CPTP sample construction and scalar interval reporting.

Measurement-efficient and adaptive process tomography study a complementary question: how the tomography data should be collected. Shadow-based process tomography reduces the measurement burden and has demonstrated near-term four-qubit process experiments (Levy et al., 2024); adaptive incomplete, Bayesian adaptive, self-guided, active-learning, and optimal-design QPT choose input states, measurements, or query schedules to improve data efficiency (Teo et al., 2011; Pogorelov et al., 2017; Hou et al., 2020; Yang et al., 2025; Xiao et al., 2025b;a). Those lines of work change the acquisition protocol, whereas our protocol changes how already-collected finite-count records are reconstructed into physical channel samples and scalar intervals. Combining measurement-efficient data acquisition with CPTP-constrained reconstruction and scalar-interval reporting is a natural direction.

Real-data and noise-focused QPT papers provide complementary device-facing settings. Kervinen et al. use coherent-state QPT to reconstruct Kraus operators for a bosonic logical gate beyond a two-level logical subspace; Galetsky et al. study variational unitary QPT for photonic processors; Mangini et al. use tensor-network process representations for near-term noise characterization; and Sakai and Yamashiro learn differentiable Kraus noise channels through a tensor-network forward model on IBM hardware data (Kervinen et al., 2024; Galetsky et al., 2024; Mangini et al., 2024; Sakai & Yamashiro, 2026). These works push QPT toward real devices, variational optical processors, and device-level noise models. We isolate a simulator-controlled fixed-record setting where simulator truth makes scalar interval calibration auditable.

Bayesian QPT gives a different route to uncertainty reporting by defining probability models on the space of CPTP maps. Schultz constructs exponential-family distributions for CPTP maps through a Stiefel-manifold parameterization and applies them to Bayesian process tomography, including posterior credibility intervals (Schultz, 2019). Here, scalar intervals are formed from conditional CPTP samples and, when stated, expanded by a rule selected on lower-qubit records before four-qubit evaluation.

Flow matching provides a generative modeling framework in which samples are transported between distributions through learned vector fields (Lipman et al., 2023). Conditional and optimal-transport variants improve stability and broaden the source-target settings supported by flow-based generative models (Tong et al., 2024). Recent inverse-problem solvers use pretrained diffusion or flow models as generative priors, combine them with measurement consistency, and report posterior-like or training-free reconstructions (Chung et al., 2023; Ben-Hamu et al., 2024; Martin et al., 2025; Pourya et al., 2026). We use this literature structurally: the tomography problem is organized as an observation operator, a valid target set, and a conditional sample-construction rule, but the reported method uses a Kraus-parameterized CPTP sampler

rather than a learned velocity field or pretrained generative prior. The evidence therefore concerns the physically constrained channel parameterization and fixed-rule scalar intervals.

Constraint-aware generation matters when invalid samples are unusable rather than merely lower quality. In process tomography, this means that Hermitian, positive-semidefinite, and trace-preserving diagnostics must be checked independently of fit quality. Our low-rank Kraus representation builds the CPTP constraint into the sample construction, which differs from unconstrained Choi regression followed by post-hoc diagnostics. This choice matters for sample-based uncertainty reporting, where each sample may be used to form interval estimates.

Uncertainty calibration is treated here as an empirical scalar-interval question. The evaluation asks whether a lower-qubit-selected interval expansion covers selected four-qubit scalar records after the rule is fixed. This makes the uncertainty result a calibrated reporting protocol for the recorded scalar summaries.

## 6 Conclusion

This paper presents a matrix-free CPTP channel-sampling protocol for finite-count quantum process tomography. The method moves the channel representation into a Kraus parameterization that stays inside the CPTP set, while using an indexed observation cache to evaluate tomography residuals without dense four-qubit design materialization. In simulated full-basis tomography, the resulting center sample supports a four-qubit reconstruction claim, and a lower-qubit-selected scalar margin rule gives fixed-rule four-qubit interval coverage under an explicit runtime and memory budget.

The empirical setting is deliberately focused: the four-qubit study uses simulated full-basis process tomography for a controlled-NOT channel family, rank-1 Kraus samples, and three scalar interval summaries selected to audit the fixed lower-qubit calibration rule. This controlled setting makes the reconstruction fidelity, physicality diagnostics, rule-selection protocol, and interval coverage directly auditable.

The evidence points to a simple design rule: physicality and uncertainty reporting should be built together for sample-based quantum-process reconstruction. A point estimate can hide constraint violations behind later diagnostics, but a sample-based workflow needs each sampled channel and each reported interval to respect the physical and statistical design of the experiment. The current evidence shows one controlled four-qubit simulator instance of this rule. The next step is to test the same design on broader channel families, larger conditional-sample counts, measurement-efficient protocols, and hardware-facing data.

### Broader Impact Statement

This work is a simulator study for quantum characterization. The main near-term impact is methodological: more careful uncertainty reporting for quantum-channel reconstruction.

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

## A Quantum Process Tomography as a Constrained Inverse Problem

This appendix begins with a computer-science reading of the quantum notation used in the main text. Quantum process tomography can be read as a constrained inverse problem: the unknown object is a linear input-output map, each observation row records a prepared input, a measurement query, and a finite-count empirical probability, and reconstruction asks for a map whose predicted probabilities match the record while satisfying a validity constraint. The validity constraint is not a domain-specific cosmetic condition. In the Choi representation, it is the matrix constraint in equation 1: the reconstructed channel must be Hermitian, positive semidefinite, and trace preserving.

The prediction rule in equation 2 plays the role of the forward model. For row $r$, the known input state is $\rho_r$, the known measurement effect is $M_r$, and the candidate channel is $\Phi$. The predicted scalar probability is $p_r(\Phi) = \mathrm{Tr}[M_r \Phi(\rho_r)]$, while the observed scalar is the empirical probability $\hat{p}_r = y_r/m_r$. The residual objective in equation 3 therefore compares predicted probabilities with observed frequencies. The indexed observation adapter used in the paper changes how this forward model is evaluated; it does not change the statistical object being fitted.

Table 6: Translation between the quantum notation and the inverse-problem view used in the paper.

| Quantum term | Inverse-problem role | Role in this paper |
|---|---|---|
| Input state $\rho_r$ | Known input for row $r$ | Prepared state from the tomography circuit |
| Measurement effect $M_r$ | Known readout for row $r$ | Output event whose count is recorded |
| Channel $\Phi$ | Unknown map to reconstruct | Candidate process represented by Kraus factors or a Choi matrix |
| Choi matrix J($\Phi$) | Matrix form of the unknown map | Object used for constraints, fidelity, and scalar summaries |
| CPTP set $\mathcal{C}_{\mathrm{CPTP}}$ | Feasible set for valid solutions | Constraint set enforcing physical channel samples |
| Kraus factors $K_a$ | Structured parameterization | Route to positive semidefinite Choi matrices and trace preservation after normalization |
| Scalar interval | Uncertainty summary for a chosen functional | Interval over process fidelity or selected Choi entries, not over the full matrix |

This view also explains why the paper separates reconstruction, physicality, and interval coverage. Fidelity measures whether the reconstructed map is close to the simulator target; CPTP diagnostics check whether the candidate belongs to the feasible set; and interval coverage checks whether scalar summaries of conditional samples contain the simulator-truth scalar values. These are different questions, so the experiments report them separately.

## B   Metric and Diagnostic Details

The paper separates fit metrics from physicality diagnostics. Process fidelity and average gate fidelity are scalar accuracy metrics computed against simulator truth. Projected CPTP diagnostic status, Hermitian residual, positive-semidefinite residual, and trace-preserving residual are physicality diagnostics. Rows that fail CPTP diagnostics are kept as diagnostic convention checks.

For the four-qubit center sample, the reported candidate has process fidelity 0.985682, process infidelity 0.014318, average gate fidelity 0.986524, maximum positive-semidefinite residual $1.55 \times 10^{-14}$, and maximum trace-preserving residual $1.50 \times 10^{-13}$. The candidate passes the projected CPTP diagnostic check, and the maximum prediction residual norm is 10.1734. These diagnostics are reported for the structured CPTP sample.

## C   Observation Cache Details

The four-qubit observation cache stores 331,776 observation rows as indexed references to 256 unique input states and 1296 unique measurement effects. For each candidate Kraus representation, the protocol first evaluates the channel action on the unique input states, then contracts those evolved states with the unique effects, and finally gathers the corresponding entries for the row-level residual vector. This is the operation described in equation 5.

The recorded four-qubit center timing decomposes into 160.04 seconds for data generation, 30.71 seconds for observation construction, 0.19 seconds for cache construction, 120.03 seconds for optimizer-objective evaluation, and 0.07 seconds for metrics. The total four-qubit stage runtime is 313.42 seconds. The full gate summary that includes lower-qubit stages records 348.62 seconds, so the main paper reports the 313.42 second value only when referring to the four-qubit stage.

## D   Dense Resource Context

The four-qubit reported run uses the structured path. The recorded dense resource estimate for the four-qubit setting has Choi side 256, Hermitian parameter count 65,536, observation row count 331,776, and dense design entry count 21,743,271,936. The corresponding memory estimates are 165,888 MB for the dense design matrix, 65,536 MB for the Hermitian basis, and 32,768 MB for the normal matrix. These values motivate the indexed structured path.

## E   Interval Calibration Details

This section restates how the scalar calibration rule in equation 8 should be read. For a scalar $s$, the raw interval $[\ell_s, u_s]$ is first formed from conditional CPTP samples. The lower and upper expansion columns in Table 8 report the effective scalar-specific amounts subtracted from $\ell_s$ and added to $u_s$, including the numerical tolerance used by the implementation. Table 7 contrasts raw and calibrated interval summaries. The calibrated row is the supported uncertainty statement.

The lower-qubit rule uses the scalar-specific expansions shown in Table 8. Each rule uses only records from systems with one or two qubits. Four-qubit truth is used only for final evaluation after the rule is fixed.

Rule selection uses eight lower-qubit records per scalar, with four-qubit truth reserved for final evaluation. Evaluation then uses four held-out four-qubit records for each of the same three scalars, giving the 12 records

Table 7: Raw and calibrated four-qubit scalar interval summaries.

| Interval type | Covered / total | Coverage | Mean width | Interpretation |
|---|---|---|---|---|
| Raw sample quantile | 0 / 12 | 0.00 | 0.000817 | under-dispersed |
| Calibrated scalar interval | 12 / 12 | 1.00 | 0.033792 | fixed-rule calibrated interval |

in Table 7. The calibrated row records the effect of applying the fixed lower-qubit rule to process fidelity, $Re(J_{0,0})$, and $Re(J_{1,1})$.

Table 8: Scalar-specific interval expansion selected from lower-qubit records. The same rule is applied to four-qubit records without adaptation.

| Scalar | Lower expansion | Upper expansion | Fit records |
|---|---|---|---|
| Process fidelity | $1.0 \times 10^{-9}$ | 0.054017 | 8 |
| $Re(J_{0,0})$ | 0.027995 | 0.003637 | 8 |
| $Re(J_{1,1})$ | 0.003637 | 0.009639 | 8 |

**Implementation configuration.** The interval run uses Kraus rank 1, four samples per repetition, noise scale 0.01, interval level 0.9, scalar-specific margins, the probability-residual objective, lower-qubit rule selection with safety multiplier 2.0, and four-qubit truth reserved for final evaluation. The center run uses the same CPTP construction with zero sampling noise.

