# OpenReview forum: "Sample-Based Constrained Inference for Matrix-Free Quantum Process Tomography"
_TMLR — Under review for TMLR_

### Review · Reviewer_TkYw · 2026-06-26

**Summary Of Contributions:**

This paper proposes a matrix-free CPTP-constrained protocol for simulated quantum process tomography, using indexed observation caches and Kraus-factor normalization to produce physical channel samples. The main evidence is a four-qubit simulated CNOT-family experiment with accurate reconstruction and calibrated scalar interval coverage, though the method is better viewed as constrained QPT.

**Additional Comments:**

AI was used only to help clarify and polish the language of this review. All opinions, assessments, and recommendations are my own.

**Audience:**

Yes

**Audience Explanation:**

While I selected “Yes” for the above question, I would like to clarify my reservation about the paper’s fit for the TMLR audience. The paper does not clearly present a machine-learning contribution. Although the authors state that the work is motivated by flow-matching inverse problems, quantum process tomography can already be naturally formulated as a constrained inverse problem, and inverse problems are a classical topic that is independent from diffusion or flow-matching models. Apart from this motivational framing, the paper appears to be primarily an implementation of constrained quantum tomography, without a clear technical connection to machine learning.

For this reason, I find it difficult to assess how broadly the machine-learning audience of TMLR would be interested in the work. Moreover, many closely related tomography papers (as I added above), including some that adapt machine-learning methods, are published in physics or quantum-information venues because the main target problem is fundamentally quantum-physical. Personally, I think this paper may be better suited to a physics or quantum-information venue than to TMLR.

**Claims And Evidence:**

No

**Claims Explanation:**

1. The related-work positioning is incomplete. It appears to miss important related work such as [1] on unsupervised-learning/tensor-network QPT, [2] on 7-qubit tensor-network QPT on hardware, [3] on projected least-squares QPT with error bounds and experiments up to 7 qubits, and [4] on Bayesian QPT with CPTP-map distributions and posterior credibility intervals.

2. The paper is missing a substantive comparison with [1]. This comparison is particularly important because [1] studies the same broad task of reconstructing quantum processes from tomographic measurement data using a structured, ML-inspired representation, and includes simulated multi-qubit QPT experiments that are close enough to serve as a meaningful baseline or at least a detailed point of comparison.

3. The main four-qubit benchmark appears too narrow and overly simplified to support a broad methodological claim. The evaluation is limited to a simulated noiseless CNOT-family unitary channel, where the chosen rank-1 Kraus representation is already well matched to the target. The paper should test more challenging settings, such as noisy channels, higher Kraus-rank/non-unitary processes, multiple circuit families, and scaling across system sizes, to show that the method is more than a controlled implementation check.

4. The dense design-matrix comparison is not very informative, since explicitly materializing the dense four-qubit design is an obvious straw-man baseline. A stronger evaluation should compare against competitive matrix-free, Kraus/Stiefel, projected, or tensor-network QPT methods, as mentioned in bullet point 2.





[1]Torlai, Giacomo, et al. "Quantum process tomography with unsupervised learning and tensor networks." Nature Communications 14.1 (2023): 2858.

[2]Dang, Aidan, et al. "Process tomography on a 7-qubit quantum processor via tensor network contraction path finding." arXiv preprint arXiv:2112.06364 (2021).

[3]Surawy-Stepney, Trystan, et al. "Projected least-squares quantum process tomography." Quantum 6 (2022): 844.

[4]Schultz, Kevin. “Exponential Families for Bayesian Quantum Process Tomography.” Physical Review A 100, 062316 (2019).

**Requested Changes:**

1. Cite and discuss the missing related works listed above, and clarify the similarities and differences relative to them.
2. Add a substantive comparison with closely related QPT methods, rather than only comparing against dense design-matrix materialization.
3. Broaden the experiments to harder settings, such as noisy/non-unitary channels, higher Kraus-rank processes, multiple channel families, and scaling across system sizes.

---

### Review · Reviewer_ENyk · 2026-07-08

**Summary Of Contributions:**

In this paper, the authors give a quantum channel reconstruction algorithm that is sample-based, instead of the usual single point estimates. Further, their algorithm avoids an explicit dense matrix materialization which would be computationally expensive. To this end, they combine a few different components: indexing basis objects, creating low-rank Kraus components to create a completely positive trace-preserving matrices (which are necessary for a physically valid channel), and finally using a scalar calibration rule that is tuned on lower qubits and applied to a higher qubit setting without adaptation (hence avoiding more expensive computations).

**Audience:**

No

**Audience Explanation:**

Currently, I do not see why this is relevant to the machine learning community. Neither the outcomes nor the techniques seem to be something that might help a machine learning researcher. However, I am not an expert in the area, and would be happy if the authors shed more light on this.

**Claims And Evidence:**

Yes

**Claims Explanation:**

The claims seem to be reasonable to me. They seem to be proposing an alternate approach to reconstructing quantum channels and support it with experimental findings. It seems to be a new line of thought, and there doesn't seem to be a benchmark to compare this to.

**Requested Changes:**

Hi, thank you for your paper. I have a few questions/suggestions.

1. Can you please shed more light on why this is relevant to the ML community?
2. I would appreciate if there was a section providing an intuitive explanation of why using your algorithm makes sense, and similarly an intuitive explanation of the results that you obtain in the experiments. PS: I am not well-versed in the area, and found it very hard to make sense of them.
3. Do you have quantitative complexity estimates of the running time etc. of your algorithm, and comparisons to anything similar in literature. Including them in the write up would be helpful.

---

### Review · Reviewer_a4xH · 2026-07-12

**Summary Of Contributions:**

**Summary:**\
This paper presents a computational pipeline for finite-count quantum process tomography that combines an indexed, matrix-free observation interface with a Kraus-factor parameterization. The former avoids materializing the full four-qubit Choi design matrix, while the latter ensures that reconstructed and sampled channels are CPTP by construction. The paper also proposes a heuristic scalar-interval calibration rule based on lower-qubit records and evaluates the pipeline on a simulated four-qubit noiseless controlled-NOT process.

**Strengths:**
1. This paper addresses a genuine memory bottleneck in full-basis quantum process tomography.
2. Kraus normalization provides a simple mechanism to ensure that every generated channel sample is physically valid.
3. The implementation details and resource usage are reported.

**Weaknesses:**
1. The main contribution is primarily an engineering integration of existing ideas (e.g., matrix-free operator evaluation, Kraus parameterization, and Stiefel/polar normalization), rather than a new inference method.
2. The authors mainly compare their proposed method against an explicitly materialized dense design matrix. However, existing QPT implementations already exploit some mechanisms to reduce computation. The paper provides no direct comparison with these realistic baselines.
3. The uncertainty samples lack a clear statistical interpretation. They are simply obtained by adding Gaussian noise to the Kraus factor.
4. The four-qubit experiment is highly favorable: the target is a noiseless unitary channel, and the model uses Kraus rank $R=1$, which restricts all reconstructed and sampled channels to be unitary. No general noisy or higher-rank four-qubit channel is tested.

**Audience:**

Yes

**Audience Explanation:**

Yes. Researchers interested in quantum tomography may find the matrix-free CPTP pipeline useful, although the appeal is likely limited by the narrow experimental setting and weak statistical validation.

**Claims And Evidence:**

No

**Claims Explanation:**

The submission provides clear evidence for its narrow engineering claims: the four-qubit implementation avoids explicit dense design-matrix materialization, produces CPTP channels by construction, achieves the reported fidelity, and stays within the stated runtime and memory budget.

However, the broader claims are not convincingly supported. E.g., the channel samples are generated using an ad hoc fixed-scale Gaussian perturbation. The nominal 90% calibration has no theoretical guarantee and is evaluated on only 12 scalar records. Moreover, the main experiment uses a rank-one model on a noiseless unitary target and is only compared with the naive implementation. There are no direct comparisons with existing Qiskit implementations.

**Requested Changes:**

Critical:
1. Add direct comparisons with realistic baselines, such as Qiskit linear inversion and constrained CVXPY fitting.
2. Evaluate non-unitary channels.
3. Clarify the methodological novelty relative to existing structured QPT implementations.
4. Statistically justify the fixed-scale Gaussian perturbation and interval expansion.

Strengthening:
1. Release source code and data-generation details to support reproducibility.

---

### Review · Reviewer_ovDZ · 2026-07-13

**Summary Of Contributions:**

The paper presents a matrix-free pipeline for finite-count quantum process tomography, combining an indexed observation interface, a CPTP-by-construction Kraus parameterization, and a heuristic calibration rule learned from lower-qubit simulations.

Strengths:
- It addresses a memory bottleneck in full-basis QPT by indexing the observation cache instead of materializing the dense design.
- The calibration rule is fixed on lower-qubit records before four-qubit truth is opened, which is good non-leaking practice.
- Runtime, memory, physicality, and coverage are reported clearly.

Weaknesses:
- The method mainly integrates existing ideas rather than introducing a new inference method.
- The Gaussian perturbation samples have no clear statistical connection to the finite-count observation model.
- The reconstruction is evaluated only in a noiseless rank-one unitary setting with zero optimizer steps, and the uncertainty evaluation uses just four samples per record.
- There is no comparison with realistic constrained or matrix-free QPT baselines.

**Audience:**

Yes

**Audience Explanation:**

Yes. Building a hard validity constraint into each sample so that sample-based uncertainty stays meaningful is relevant beyond QPT to constrained inverse problems, though broader interest depends on stronger statistical validation.

**Claims And Evidence:**

No

**Claims Explanation:**

The implementation claims are reasonably supported.
The uncertainty evidence, however, is the weak point: four samples cannot support a nominal 90% interval, and the 12/12 coverage reduces to three related statistics on just four records.
The calibration margins are also transferred from one- and two-qubit systems to four qubits without justification.
Separately, the reconstruction covers only one favorable unitary setting and never runs the stated optimizer, so it evaluates the adjoint-spectral initializer rather than the full method.

**Requested Changes:**

- Run the stated optimizer, or state that the reported reconstruction is only the adjoint-spectral initializer.
- Report coverage from many more samples and over multiple independent datasets, not four samples on one record set.
- Give the perturbation and intervals a statistical basis, or relabel them as heuristic sensitivity intervals.
- Extend beyond the noiseless unitary case to noisy, nonunitary, and higher-rank channels.
- Benchmark against real QPT methods (Qiskit inversion, CVXPY constrained fit, or a matrix-free method like GD-QPT), not only the dense design.
- Add ablations over sample count, Kraus rank, perturbation scale, and optimizer steps, and release code.